# Application of Flow Cytometry Using Advanced Chromatin Analyses for Assessing Changes in Sperm Structure and DNA Integrity in a Porcine Model

**DOI:** 10.3390/ijms25041953

**Published:** 2024-02-06

**Authors:** Estíbaliz Lacalle, Estela Fernández-Alegre, Belén Gómez-Giménez, Manuel Álvarez-Rodríguez, Beatriz Martín-Fernández, Cristina Soriano-Úbeda, Felipe Martínez-Pastor

**Affiliations:** 1Institute of Animal Health and Cattle Development (INDEGSAL), University of León, 24071 León, Spain; elacalle@bianorbiotech.es (E.L.); bmarf@unileon.es (B.M.-F.); 2Bianor Biotech SL, 24071 León, Spain; 3Department of Animal Reproduction, National Institute for Agricultural and Food Research and Technology, Spanish Scientific Research Council (INIA-CSIC), 28040 Madrid, Spain; manuel.alvarez@inia.csic.es; 4Department of Molecular Biology (Cell Biology), University of León, 24071 León, Spain; 5Department of Medicine, Surgery and Veterinary Anatomy (Animal Medicine and Surgery), University of León, 24071 León, Spain; c.soriano.ubeda@unileon.es

**Keywords:** boar, spermatozoa, cooled storage, flow cytometry, chromatin status, oxidative stress, DNA fragmentation, sperm functionality, multivariate analysis

## Abstract

Chromatin status is critical for sperm fertility and reflects spermatogenic success. We tested a multivariate approach for studying pig sperm chromatin structure to capture its complexity with a set of quick and simple techniques, going beyond the usual assessment of DNA damage. Sperm doses from 36 boars (3 ejaculates/boar) were stored at 17 °C and analyzed on days 0 and 11. Analyses were: CASA (motility) and flow cytometry to assess sperm functionality and chromatin structure by SCSA (%DFI, DNA fragmentation; %HDS, chromatin maturity), monobromobimane (mBBr, tiol status/disulfide bridges between protamines), chromomycin A3 (CMA3, protamination), and 8-hydroxy-2′-deoxyguanosine (8-oxo-dG, DNA oxidative damage). Data were analyzed using linear models for the effects of boar and storage, correlations, and multivariate analysis as hierarchical clustering and principal component analysis (PCA). Storage reduced sperm quality parameters, mainly motility, with no critical oxidative stress increases, while chromatin status worsened slightly (%DFI and 8-oxo-dG increased while mBBr MFI—median fluorescence intensity—and disulfide bridge levels decreased). Boar significantly affected most chromatin variables except CMA3; storage also affected most variables except %HDS. At day 0, sperm chromatin variables clustered closely, except for CMA3, and %HDS and 8-oxo-dG correlated with many variables (notably, mBBr). After storage, the relation between %HDS and 8-oxo-dG remained, but correlations among other variables disappeared, and mBBr variables clustered separately. The PCA suggested a considerable influence of mBBr on sample variance, especially regarding storage, with SCSA and 8-oxo-dG affecting between-sample variability. Overall, CMA3 was the least informative, in contrast with results in other species. The combination of DNA fragmentation, DNA oxidation, chromatin compaction, and tiol status seems a good candidate for obtaining a complete picture of pig sperm nucleus status. It raises many questions for future molecular studies and deserves further research to establish its usefulness as a fertility predictor in multivariate models. The usefulness of CMA3 should be clarified.

## 1. Introduction

The detection of biomarkers to predict sperm quality and functional integrity has gained much interest over the last decade for human health and animal production. However, routine spermiograms have largely been unsuccessful in attaining this aim [1], though techniques based on molecular approaches show promise [2,3,4]. In this regard, sperm chromatin evaluation has recently entered the set of available techniques for studying fertility as a biomarker of sperm fertility [5,6]. In this regard, sperm DNA fragmentation (SDF) is used most in human clinics and breeding centers [7,8]. However, SDF is only one of many levels in sperm chromatin evaluation, and solely focusing on it ignores the complex structural, epigenetic, and other molecular features affecting sperm fertility, embryo development, and even the health and well-being of the offspring [9,10].

During late spermatogenesis, the haploid complement in the sperm nucleus is reorganized as protamines replace most histones [11]. Moreover, during epididymal maturation, a network of disulfide bridges among protamines further compacts it [12]. All these events protect the integrity of sperm DNA and have repercussions on the embryo’s gene expression and the fetus’s and newborn’s development [13].

Sperm chromatin compaction is crucial for pig spermatozoa due to its peculiar structure, making it unique among other domestic species [8,14]. The pig only expresses the PRM1 gene as the P1 protein, which also contains ten cysteines (five to seven in most mammals), resulting in many disulfide bridges [14]. It produces a very compact sperm nucleus that yields relatively low SDF levels. Therefore, the threshold for considering a sample positive for DNA damage is much lower than in human or cattle [15,16,17], and evaluating DNA damage in this species is challenging [18]. Notwithstanding, many studies have associated SDF in pigs with decreasing prolificacy (possibly due to failure in embryonic development [19]), often after prolonged storage [8,12,14,15,16,17,20].

Chromatin compaction can be quickly evaluated by fluorochrome-exclusion methods using intercalating fluorochromes (higher compaction hampers access to the tightly packed sperm DNA, resulting in a lower fluorescence). This is the principle behind the %HDS (high DNA stainability) of the SCSA (sperm DNA structure assay), which depends on the access of acridine orange to the sperm nucleus [21]. Chromomycin A3 (CMA3) analysis is based on a similar principle [22], competing with protamines to bind DNA in a Mg^2+^-dependent process [23]. Widely used in humans, high CMA3 labeling has been associated with lower male fertility [24]. Nevertheless, the interpretation of the CMA3 assay is complex since it is affected by other factors, such as DNA fragmentation and the overall quantity of disulfide bridges among protamines [25].

Indeed, the formation of disulfide bridges is also a critical parameter for the compaction of the sperm nucleus. It occurs as part of sperm maturation in the epididymis, mediated by nuclear glutathione peroxidase 4 (nGPX4) [26]. It is also a dynamic process, reversed at fertilization and affected by the sperm environment [27,28,29,30]. Monobromobimane (mBBr), a fluorescent reagent that covalently binds to free sulfhydryl groups, can evaluate the level of disulfide bridges in the sperm nucleus [31].

Additionally, oxidative stress can affect DNA integrity. Spermatozoa are especially sensitive to oxidative stress, which produces lipid peroxidation and oxidation of DNA nitrogenous bases [32], a major cause of SDF. Although not much has been explored on this topic [33], DNA oxidation could be relevant for boar sperm analysis because of the low levels of detectable SDF typical of this species. In this regard, deoxyguanosine is considered a preferential target for oxidative insult, and therefore, immunodetection of 8-oxo-dG is the preferred method to estimate DNA oxidation levels, including in spermatozoa [34].

Besides SDF, studying these parameters could be relevant for the pig, a species of economic interest and great value for research due to its peculiar sperm chromatin structure. Therefore, our first objective was to evaluate a range of sperm chromatin parameters in samples from a boar breeding center, focusing on variability among males and along a prolonged storage time. We hypothesized that intrinsic (boar) and extrinsic (storage) factors could affect DNA insult (as SDF and oxidation) and chromatin structure (as compaction, protamination, and disulfide bridges) differently. Our second objective was to enhance the study of these many variables using multivariate analyses. These statistical methods enable us to test the effects and associations of multiple explanatory and response variables simultaneously, strengthening the information obtained from analyzing numerous cellular and molecular features [29,35].

To this end, we applied techniques that have been little tested in pig spermatozoa, and the methodology used flow cytometry for a quick, sensitive, and objective analysis and streamlined sample storage and workflow, enabling the simultaneous analysis of many samples.

## 2. Results

### 2.1. Results for the Clustering of Motility Data

Motility data provides plenty of information which is partly lost when obtaining averages of kinematic variables. Data clustering enables the obtention of subpopulations with specific swimming patterns, enabling a more in-depth sperm motility analysis. The cluster analysis on the raw CASA data produced three subpopulations, described in Table 1. Two were termed Slow and Fast, respectively, considering the relative median velocities (ALH and BCF, related to motility vigor, were also high in the Fast subpopulation). The third one was termed Hyper, since its motility parameters (fast, low linearity and straightness, and high ALH and dance) resembled hyperactivated spermatozoa.

### 2.2. Extensive Modifications of Sperm Chromatin Are Evident after Long Refrigerated Storage, with a Relevant Boar Effect

Sperm storage changes the quality of the doses, including the chromatin. Whereas previous studies have focused on DNA fragmentation [16], with few changes, we have focused on the overall chromatin structure, finding that a long storage period significantly affected many parameters. Taking sperm physiology into account, Appendix A shows the average results at days 0 and 11 for sperm motility and physiology parameters, including the *p* values from the linear mixed-effect models, with Day, Boar, and their interaction as fixed factors (full ANOVA results in Appendix A). All motility parameters noticeably decreased, and only the percentage of the slow subpopulation increased after 11 days of storage, while parameters related to sperm viability, membrane structure, and mitochondrial activity showed much smaller changes (albeit highly significant ones in most cases). Additionally, ROS production, both as cytoplasmic peroxide and mitochondrial ROS, increased at day 11. Overall, sperm structural features (especially the plasma membrane) resisted long storage well, with a more evident increase of oxidative stress, features essential for interpreting chromatin changes.

Chromatin parameters (Table 2; full ANOVA results in Appendix A) were affected differently by storage. The SCSA variables remained similar. Still, %DFI (DNA fragmentation) increased significantly (D0: 1.12% ± 0.07 vs. D11:1.41% ± 0.07; *p* = 0.003); in any case, they were well below fertility-compromising levels. The related variable on the level of oxidation (8-oxo-dG MFI) showed a significant oxidation increase due to storage (D0: 2.54 ± 0.05 vs. D11: 2.80 ± 0.05; *p* < 0.001). The evaluation of disulfide bridges indicated a decrease of those after the storage period (disulfide levels, D0: 33.28 ± 0.31 vs. D11: 21.31 ± 0.31; *p* < 0.001), congruent with an increase in median fluorescent intensity (MFI) mBBr (D0: 0.95 ± 0.01 vs. D11: 1.18 ± 0.01; *p* < 0.001). Finally, the nuclear compaction assessed by CMA3 decreased slightly (increased fluorescence), indicating higher probe accessibility and a low protamination level (CMA3 MFI, D0: 1.083 ± 0.004 vs. D11: 1.099 ± 0.004; *p* = 0.002), with a corresponding significant increase in the highly fluorescent subpopulation (high CMA3), which refers to the subpopulation of spermatozoa with less DNA compaction (D0: 6.45 ± 0.24 vs. D11: 7.51 ± 0.24; *p* < 0.001).

### 2.3. Chromatin Structure Variables Are Related to Different Nuclear Features

The correlation analysis evidenced some differences in the association between the chromatin parameters after semen collection (D0, Figure 1a) and after 11 days of storage (Figure 1b). Significant correlations within each technique are expected due to the relationship among these variables, and they are described elsewhere. Considering D0, SD-DFI, and %DFI (SCSA variables related to DNA damage) correlated only between each other, with %HDS and SD-DFI with the level of oxidized DNA (anti-8-oxo-dG MFI), with some association of SD-DFI with mBBr outcomes. However, %HDS, a proxy of chromatin maturation or compactness, correlated with mBBr variables (overall, a negative relationship with disulfide bridges), with some positive association with the presence of low-CMA3 events and oxidative damage. The mBBr variables correlated among themselves, and (apart from %HDS) the levels of free tiols (mBBr MFI) and disulfide bridges (disulfides) were positively and negatively correlated with the DNA oxidation levels, respectively. Interestingly, CMA3 variables did not correlate with the other techniques; only low CMA3 had a slight positive correlation with %HDS. CMA3 parameters only had correlations amongst themselves.

The number of significant correlations decreased in general, possibly due to the different change rates of the chromatin variables during storage. We cannot discard the effect of the boars’ variability, as indicated in Section 2.2. The most relevant change was the significant correlations of %DFI with the low and medium-mBBr populations (positively and negatively, respectively). Also, the negative correlation between the populations of CMA (high CMA3 and low CMA3) was stronger after 11 days of storage.

The Appendix A shows the complete correlation study (Appendix A). Variables related to better quality (e.g., motility, velocity, viability, mitochondrial activity) were associated with better chromatin status, with the contrary for some quality variables related to worse condition (notably, acrosomal status). The associations slightly changed with storage, but not dramatically, and most correlations were relatively low. Interestingly, 8-oxodG levels were associated with many CASA and cytometry variables at D0. On D11, fewer correlations were significant due to the sample divergence. Nevertheless, the ratio of viable spermatozoa with high mitochondrial superoxide production stood out as a significant positive correlation.

### 2.4. Hierarchical Clustering and Principal Components Analysis Unveils Relationships among Chromatin Variables and with Sperm Motility and Physiology Variables

Correlation analysis is a valuable technique for identifying putative relationships among variables, but the interpretation is often difficult when the number of variables is large. Here, multivariate analyses such as variable clustering and principal component analysis help to describe the data and clarify such associations [29,36]. Variable clustering (Figure 2) highlighted the change in relationships among different variables at the two analysis times (D0 and D11). Specifically, the mBBr variables and the 8-oxo-dG MFI varied considerably in this analysis. At D0 (Figure 2a), the variables related to a lower tiol availability (low and moderate mBBr fluorescence) were more directly related to the SCSA.

In contrast, the variables related to a higher tiol availability and disulfide bridge index were associated with the greatest amount of oxidative damage to the DNA (8-oxo-dG). Again, CMA3 variables were only related between themselves. At D11 (Figure 2b), SCSA variables, mainly %HDS, were more associated with 8-oxo-dG amounts. The mBBr populations tended to form a cluster, and the CMA3 variables, clustered aside by D0, were associated with disulfide bridge quantity.

The clustering of the sperm quality variables (Appendix A) revealed interesting associations of the chromatin variables with sperm motility and physiology. In the D0 analysis (Appendix A), whereas CMA3 variables clustered near the motility ones, SCSA and 8-oxodG clustered with those related to mitochondrial activity. The mBBr variables instead were split among three clusters: the mitochondrial variables (disulfide bridge levels and MFI), with viability, acrosomal status, and mitochondrial ROS (low/moderate mBBr); and with the remaining physiological variables (high mBBr). This pattern radically changed in the second analysis at D11 (Appendix A), with motility and physiological variables clustered together and apart from the chromatin ones. The only exceptions were the amount of cytoplasmic ROS clustering with the mBBr subpopulations (near the SCSA and 8-oxo-dG variables) and the Hyper subpopulation, clustered among the latter.

Principal component analysis (PCA) is a descriptive analysis that yields a series of variables (principal components) as linear combinations of the original variables. The principal components (PC) attempt to capture the variability in the data. Thus, the first PC explains the most variance, then the second, and subsequently, any lower variance. As in the previous analyses, we first focused on the chromatin variables. In this case, the first two PCs explained 31.7% and 15.5% of the variance (Appendix A). While the first four PCs explained nearly 80% of the variance and could be retained for a full analysis, we focused on this first PCA for simplification, and because the PC1 enabled a good separation between days (Appendix A shows the details for the other PC combinations). Figure 3 shows the PCA results for these two PCs, with the left panel displaying the projection of the individual observations projected in the space defined by the two first PCAs and the right panel displaying the variable loadings for each PC (Table 3 details the loadings and contributions of the original variables). The analysis day was the main factor separating observations (PC1, influenced by the mBBr variables). This separation was confirmed by hierarchical clustering in the PCA data, yielding two clusters that grouped most of the D0 observations in the lowest range of PC1 and most of the D11 observations in its highest range. The SCSA and 8-oxo-dG variables showed similar trends and were nearly orthogonal to the mBBr variables (influencing PC2) and associated with between-male variability. CMA3 variables were intermediate and influenced both PCs to different degrees, showing a mixed influence from day and boar.

A second PCA included all the variables from the analyses (Appendix A). To keep the interpretation simple, only the first two PCs were considered (due to the large number of variables entered, at least eight PCs might be retained for a complete analysis, as Appendix A shows). In this case, due to the large initial number of variables, the clustering was less defined, but it roughly discriminated between D0 and D11 again (Appendix A), and mainly along the PC1 axis, with PC2 relating to between-boar variability. This distribution, even if less defined than in the previous analysis, evidenced some relationships between the chromatin status variables, motility, and physiology. In this joint analysis, PC1 was influenced mainly by motility and mBBr variables, mitochondrial ROS, and 8-oxo-dG (Appendix A); and PC2 by the ratios of apoptotic and capacitated spermatozoa, the proportion of hyperactivated spermatozoa, and to a lesser degree, the CMA3 variables (loadings and contributions detailed in Appendix A). The remaining variables presented a mixed influence on both PCs in this projection.

The SCSA and 8-oxo-dG parameters showed a similar trend, equivalently influenced by PC1 and PC2, although SD-DFI is more closely related to PC1; %DFI and 8-oxo-dG show a practically overlapping trend. The mBBr parameters, especially disulfide bridges and mBBr MFI, were much more related to PC2.

## 3. Discussion

### 3.1. Analysis of the Chromatin Structure in the Pig Model

We propose a more comprehensive evaluation of the sperm chromatin in the porcine species beyond the generally used DNA fragmentation tests. This approach could be practical in pig spermatozoa due to its particular sperm chromatin structure, which is more compact than that of other species. Pig and cattle sperm nuclei only contain protamine 1, unlike sperm in humans, rodents, and stallions, which also present protamine 2, which achieves a lower compaction and possibly a more vulnerable DNA [37]. Moreover, protamine 1 in pigs contains a comparatively higher number of cysteine residues, favoring the formation of a denser disulfide network [14].

This high compaction could also be the reason for the low SDF levels and normality threshold in this species. While SDF levels from 10% to 20% are expected in other species and not considered worrying (e.g., in human and cattle), measurements in pigs are usually below 5% [6]. Therefore, SDF has a lower discriminant power than other species due to its smaller effect [18,38]. Thus, more extensive panels could improve the evaluation of boars, helping to detect both subfertile individuals and transitory decreases in sperm quality.

As a result, the analysis of chromatin maturity, compaction, and disulfide bridges has been used more sparingly in this species [29,30,39,40]. The present study proposes a combination of the SCSA, CMA3, and mBBr techniques to achieve a better understanding of the overall structure of the sperm nucleus.

### 3.2. Detection of 8-oxo-dG as a Marker of DNA Oxidative Insult and a Proxy for SDF

This is the first report on using single-cell 8-oxo-dG detection for porcine spermatozoa. The high compaction of the sperm nucleus seems to be a challenge when performing immunodetection in this species. Indeed, the optimal detection of 8-oxo-dG was only achieved in our laboratory after pre-treatment with 10 mM DTT and 0.1% Triton X-100, whereas a similar assay in sheep required only 2 mM DTT and 0.5% Triton X-100 [41]. Similarly, other authors have indicated that boar spermatozoa should be treated with 5 mM DTT and 1 M NaCl to detect DNA damage when using a TUNEL assay [40].

Our results concerning DNA oxidative damage must be interpreted considering the flow cytometry assessment of oxidative species. Interestingly, cytoplasmic and mitochondrial ROS levels were low (as we observed previously in unstimulated samples [36,42]), and only the latter were significantly affected by boar and storage. Therefore, it is not surprising that the correlations of these variables with the chromatin variables were non-significant or of little relevance. These results align with recent work linking sperm DNA fragmentation in this species with other factors, independent of oxidative damage [43]. Nevertheless, many studies have shown that pig spermatozoa could show higher levels of oxidative stress and DNA damage if obtained from stressed animals (e.g., heat stress [44]) or if submitted to harmful treatments, such as cryopreservation [45,46] or bacterial contamination [42].

Therefore, the levels of DNA oxidation in this study might emerge not from sperm oxidative stress but from the maturation process in the epididymis. This maturation includes the oxidation of thiols into disulfides [47], a highly regulated process but one involving ROS generation, which might affect DNA. Boar spermatozoa are highly susceptible to oxidative damage because of the high proportion of polyunsaturated fatty acids in their membranes and low cytosolic antioxidant defenses [48]. Therefore, even low levels of ROS could initiate oxidative cascades, resulting in basal levels of 8-oxo-dG.

In any case, 8-oxo-dG results in the present study indicated that oxidative damage to pig sperm DNA, even if low, was affected by the boar and increased with time in storage. More interestingly, at D0, it correlated with other chromatin parameters, notably with SCSA and mBBr (higher fragmentation, lower compaction, and fewer disulfide bridges). These results are associated with findings in other species, such as humans, where 8-oxo-dG formation and DNA fragmentation are correlated [34]. Following this reasoning, we hypothesize that the detected DNA oxidation at D0 and D11 might have emerged due to the accumulation of mild oxidative events rather than acute oxidative stress (as this might be the case in other studies, resulting in relevant SDF [49,50,51]). Interestingly, lower chromatin compaction and lower levels of disulfide bridges were associated with higher levels of 8-oxo-dG. This suggests that imperfect compaction could facilitate the access of oxidative species to the DNA.

These hypotheses could be appropriately tested in models for reproductive failure. Indeed, Viñolas-Vergés et al. [43] reported that defects in the formation of disulfide bridges could have a pivotal role in DNA integrity and fertility in the pig, and heat stress is a known inducer of SDF and likely imperfect chromatin compaction [49]. Evaluation of 8-oxo-dG levels might detect these alterations before they reflect relevant chromatin damage.

### 3.3. CMA3 for the Evaluation of Chromatin Compaction in Pig Spermatozoa

CMA3 has been used to analyze sperm compaction in other species [52,53]. However, it was less informative in our study, in line with previous reports for pig spermatozoa [29,30,43]. A possible reason could be that this technique could help identify boars with altered spermatogenesis, whereas those used in this study were subjected to strict evaluation. Therefore, males with inferior reproductive outcomes were continuously removed, possibly resulting in more homogenous results when evaluating parameters related to critical steps like protamination. However, CMA3 has proven helpful for assessing pig spermatozoa in some situations [39], and it could be applied as a way to discard suboptimal males when they enter stud centers or to evaluate the impact of conditions such as heat stress [54]

Additionally, CMA3 interpretation is controversial, since studies in human have shown a striking coincidence of DNA oxidation, SDF, and CMA3 results [34]. Indeed, CMA3 cannot reflect the degree of protamination [25], and SDF could modulate the interaction of CMA3 with DNA, altering the evaluation of chromatin compaction by this technique. However, we remain ignorant of whether the situation could be the same for pigs, considering the differences in chromatin structure for both species (including the presence of two protamines in human sperm).

### 3.4. Multivariate Analyses Enhance Discovering Variable Associations and Potential New Research Avenues

A notable result in our study was the rearrangement of associations among chromatin variables between D0 and D11, evidenced by the correlation and variable clustering studies. The storage increased the clustering of 8-oxo-dG with SCSA variables, whereas those of mBBr tended to form clusters isolated from other techniques. Therefore, some parameters (SCSA and 8-oxo-dG) could be mechanistically associated with diverging disulfide bridge dynamics. Thus, the relationship between chromatin parameters in the sperm nucleus might be a critical aspect to examine when determining the resilience and fertility of pig spermatozoa following refrigerated storage.

Indeed, the PCA suggested that the dynamics of disulfide bridges could be a relevant event during refrigerated storage, influencing the first PC (capturing the most significant part of the variance). In contrast, the influence of the SCSA or the 8-oxo-dG levels was much lower (also shown in the models’ results). The ability of mBBr analysis to find differences between treatments was not evident in previous studies [29,30] but still showed good sensitivity to incubation or storage. In any case, mBBr should be tested for discrimination power among boars and to determine if it is related to stressful events or male fertility, as is the case for SDF [15,16,38,55,56].

The PCA also demonstrated the complementarity of 8-oxo-dG levels, SDF, and nuclear maturity (%HDS), which was almost orthogonal to mBBr variables. Additionally, these variables captured much boar variability at both sampling times. Therefore, they might provide specific information on DNA status, contrasting with the structural information provided by mBBr. In this regard, %HDS is peculiar, since it was not affected by storage but by the male; our previous studies showed good sensitivity for discriminating among males and treatments and very dynamic behavior during storage and incubation [29,46]. While %HDS has been related to lower fertility in human and cattle [57,58], there is little information on the porcine species, deserving further attention.

### 3.5. Implications for Animal Breeding and the Human Clinic

We have considered a wide range of chromatin variables, including motility and flow cytometry parameters, and focused on a multivariate analysis. Our workflow in the present study included a fast preparation and easy storage of the samples, their processing in 96-well plates, and analysis in a flow cytometer provided with a plate reader, enabling fast and accurate determination of many parameters from each semen sample. This methodology is still too complex and time-consuming for the routine production of AI doses. Still, it could be incorporated into the regular evaluation of boars, especially for detecting those with a high risk of subfertility, or assessing the impact of critical periods (like summer [49]). In addition, while flow cytometers are not common in stud centers yet, this equipment is increasingly affordable and user-friendly.

Moreover, our approach goes beyond the improvement of pig breeding. This species is a valuable model for human medicine, including reproductive research [59,60]. As such, the study of human spermatozoa could benefit from applying the multivariate panel to assess sperm chromatin. SDF is still the predominant test performed in human clinics despite extensive research indicating that other evaluations (like those used in the present study) could provide valuable information on the diagnosis of infertility [7]. While the chromatin structure is very different between the two species, it would be worthy of testing if the associations between SDF, chromatin maturity, base oxidation, and disulfide bridges resemble those found for pig spermatozoa and how they vary not only between fertile and infertile patients but also at different stages during IVF procedures. We have to highlight that a subsequent step in our research is associating our findings with the fertility and prolificacy data provided by breeder farms. An advantage of using stud pigs as models in this research is the availability of fertility data for each male from hundreds of inseminated sows within months (which is impossible in human clinical research). These data would clarify the results of the multivariate analysis presented here and possibly help interpret similar analyses on human spermatozoa.

## 4. Materials and Methods

### 4.1. Solutions and Reagents

Unless otherwise indicated, all chemicals and reagents were purchased from Sigma-Aldrich, Inc. (Merck KGaA, Darmstadt, Germany), and fluorescence reagents from ThermoFisher Scientific (Waltham, MA, USA).

Phosphate buffered saline (PBS) solution for cell washing was composed of 140 mM NaCl, 15 mM KCl, 7 mM Na_2_HPO_4_, and 1.5 mM KH_2_PO_4_ (pH 7.2), with 0.5% bovine serum albumin (BSA) added when needed. Beltsville Thawing Solution (BTS) for sperm dilution was composed of 205.4 mM glucose, 20.4 mM sodium citrate tribasic, 14.9 mM NaHCO_3_, 3.36 mM disodium EDTA, and 10.1 mM KCl, and 0.5% BSA (pH 7.2).

The stocks of fluorescent probes were prepared in water (Hoechst stains, propidium iodide, PNA-Alexa Fluor 647) or dimethyl sulfoxide (DMSO) at the following concentrations: 1.8 mM Hoechst 33342 (H342), 9.4 mM Hoechst 33258 (H258), 75.0 µM YO-PRO-1 (YP), 1.0 mM 5-(and-6)-chloromethyl-2′,7′-dichlorodihydrofluorescein diacetate, acetyl ester (CM-H_2_DCFDA; CFDA), 1.0 mM merocyanine 540 (M540), 1.5 mM propidium iodide (PI), 1.0 mM peanut agglutinin (lectin from *Arachis hypogaea*) conjugated with Alexa Fluor 647 (PNA), 1.0 mM MitoTracker deep red (MTdr), 1.0 mM MitoSOX (MX), 10 mM chromomycin A3 (CMA3), and 50 mM monobromobimane (mBBr).

The antibody solution for detecting oxidized deoxyguanosine was prepared at 1:150 in PBS-0.5% BSA with the 8-hydroxy-2′-deoxyguanosine antibody (anti-8-OH-dG) conjugated to fluorescein isothiocyanate (FITC Anti-DNA/RNA Damage antibody [15A3]; ab183393; Abcam, Cambridge, UK).

### 4.2. Animals and Samples Collection

The insemination center AIM Ibérica/Topigs–Norsvin Spain (Campo de Villavidel, León, Spain) donated commercial artificial insemination (AI) doses from 36 boars (Large White and Landrace). These males were part of the center’s routine production of commercial semen doses for AI in breeders’ farms, providing semen regularly (three times every two weeks); they were under continuous evaluation by AIM, controlling physical, physiological, and semen parameters. Sows in breeding farms were inseminated with doses from the boars used in the study once, with no direct contact with boars, confirming farrowing 110–118 days after insemination. Topigs–Norsvin evaluated the fertility data from the breeders’ farm, confirming that the boars complied with the stud center’s requirements of fertility (higher than 85%) and piglets born alive per sow (higher than 12). The animals were between 12 and 22 months old and kept under standardized environmental conditions: housing with regulated temperature between 18 °C and 23 °C and 12 h light/darkness cycles. Semen was collected in consecutive weeks by the gloved-hand method. The rich fraction of every ejaculate was collected separately, filtered, and extended to up to 80 mL in a commercial semen extender for porcine AI doses (Biosem Plus, extra-long-term extender, Magapor S.L., Ejea de los Caballeros, Spain). AI doses contained 30 × 10^6^ spermatozoa mL^−1^ and were kept at 17 °C until analysis.

### 4.3. Experimental Design

Three AI doses per boar (36 boars) were used in this study. Once in the laboratory, an aliquot of each dose was processed and analyzed immediately (day 0 of the study, D0), and the remaining sample was maintained at 17 °C for 11 days and assessed (D11). These two analysis points were selected taking into account previous results [29,42,51] (good maintenance of sperm viability) and maximizing changes in the chromatin structure during the refrigerated storage (since this study also aims to select the most informative parameters; more specific time points might be assessed in subsequent studies). Analyses consisted of assessing spermatozoa functionality by flow cytometry, motility by CASA, and study of the chromatin as oxidative DNA damage, DNA fragmentation and chromatin maturity (SCSA), disulfide bonds between protamines, and chromatin protamination/condensation. For the chromatin analysis, the samples were diluted in Tris-NaCl-EDTA (TNE) buffered medium (10 mM Tris-HCl, 0.15 M NaCl, and 0.1 mM EDTA; pH 7.4) and stored at −80 °C until analysis.

### 4.4. Sperm Functionality Assessment

The analysis of spermatozoa functionality in fresh samples was carried out by flow cytometry. The probes for the analysis were prepared in BTS-BSA as previously described by our group [30] with minor modifications: viability, apoptosis, capacitation, and acrosomal status by combining 2.7 µM H342, 37.5 nM YP, 0.7 µM M540, 2.25 µM PI, and 1.5 µM PNA; and viability, cytoplasmic and mitochondrial ROS, and mitochondrial activity by combining 4.7 µM H258, 1.5 µM CFDA, 0.7 µM MX, and 300 nM MTdr. The sperm samples were diluted to 10^6^ mL^−1^ and incubated in the dark for 15 min at 37 °C.

### 4.5. Sperm Motility Analysis

A 5 µL aliquot of a fresh sample was evaluated in a pre-warmed 20 µm chamber (Magapor, Ejea de los Caballeros, Spain) on a heated stage at 37 °C. The computer-assisted sperm analysis system (CASA) consisted of a Nikon E600 microscope (Nikon, Tokyo, Japan) with ×10 negative phase contrast optics, a Basler A312fs digital camera (Basler Vision Technologies, Ahrensburg, Germany) working at 53 fps, and the ISAS v.1.19 software (Proiser, Valencia, Spain). At least 200 spermatozoa in 5 different fields were captured for each sample. The CASA provided the proportion of motile (MOT, %) and progressive spermatozoa (PROG, %, VCL > 25 µm/s and STR > 45%) per sample, and the kinematic parameters curvilinear velocity (VCL, μm/s), straight path velocity (VSL, μm/s), average path velocity (VAP, μm/s), LIN (linearity, %), straightness (STR; %), wobble (WOB, %), amplitude of the lateral displacement of the sperm head (ALH, μm), frequency of the flagellar beat (BCF; Hz), Dance (DNC, µm^2^/s), and mean Dance (DNCm, µm). Motility sperm subpopulations were obtained by processing the raw CASA data using a double clustering process described previously [61].

### 4.6. Oxidative DNA Damage Analysis

Oxidized guanine adducts indicate oxidative damage in the DNA before DNA fragmentation occurs. Samples were thawed, and 100 µL were pipetted on 96-well dishes and washed with PBS (1000× *g*, 11 min, 4 °C). The samples were fixed for 20 min with 4% formaldehyde in PBS at room temperature, then washed with PBS-0.5% BSA and incubated with 10 mM dithiothreitol (DTT) in PBS (15 min, 37 °C). After washing, spermatozoa were permeabilized with 0.1% Triton in 0.1% sodium citrate solution (30 min on ice), washed again, and incubated with the anti-8-oxo-dG antibody solution for 1 h at room temperature and protected from light. Samples were washed and incubated for at least 30 min with a 2.7 µM H342 solution in PBS at room temperature to stain the nucleus.

### 4.7. Sperm Chromatin Structure Assay (SCSA)

SCSA was carried out as previously described [62]. This assay uses acid denaturation of the sperm DNA and the intercalating acridine orange (AO) fluorochrome for assessing DNA fragmentation and compaction. AO shifts from green to red depending on whether it is bound to double-stranded DNA (dsDNA) or single-stranded DNA (dsDNA, formed from nicked DNA after denaturation), respectively. Samples were thawed, and 100 µL incubated for 30 s on ice with 200 µL of acid–detergent solution (0.08 M HCl, 0.15 M NaCl, and 0.1% Triton X-100; pH 1.2). Then, 600 µL of AO solution (6 µg/mL chromatographically purified AO in 0.1 M citric acid, 0.2 M Na_2_HPO_4_, 1 mM EDTA, 0.15 M NaCl; pH 6.0) were added to the tubes were analyzed before 3 min.

### 4.8. Chromatin Disulfide Bridge Assessment

Disulfide bond analysis was carried out with monobromobimane (mBBr), which covalently binds free thiols, enabling the quantitative measurement of reduced disulfide bonds [31]. Samples were thawed, and 100 µL was pipetted onto two 96-well plates and washed with PBS (1000× *g*, 11 min, 4 °C). Samples were resuspended with 100 µL of PBS in one of the plates and with 1 mM of dithiothreitol (DTT) in PBS to generate a set of reference samples with decondensed chromatin. After 10 min at 37 °C, the plates were washed with PBS, and 100 µL of mBBr (500 µM in PBS) were added, washed after 10 min at 37 °C, and incubated for 15 min with 2 µM PI in PBS to stain the nucleus.

### 4.9. Chromatin Condensation Assay

The level of chromatin condensation was evaluated with fluorescent antibiotic chromomycin A3 (CMA3), which bound DNA between guanine-cytosine (G-C) pairs. It indirectly measures the protamination in spermatozoa [23], since CMA3 competes with protamines for chromatin binding. In the case of high chromatin compaction related to a high level of protamination, CMA3 cannot bind its target. One hundred µL of thawed samples was displayed on 96-well dishes and washed with PBS, centrifuging 1000× *g* (11 min, 4 °C) and removing the supernatant. Samples were incubated with 100 µL of CMA3 solution (50 µM in PBS with 5 mM MgCl_2_, freshly prepared from the stock) for 20 min at 25 °C and protected from light. Samples were washed by adding PBS, centrifuging in the same conditions, and removing the supernatant. Samples were incubated for 15 min with 2.7 µM H342 in PBS to stain the nucleus.

### 4.10. Flow Cytometry Analyses

Flow cytometry analyses were carried out as detailed in previous works [29,42,51] A MACSQuant Analyzer 10 flow cytometer equipped with a 96-well plate processor and three lasers (violet at 405 nm, blue at 488 nm, and red at 635 nm) and controlled with MACSQuantify™ software v. 2.13 (Miltenyi Biotec, Bergisch Gladbach, Germany) was used for all the analyses. Spermatozoa were gated using appropriate regions using FSC/SSC (forward/side scatter), and different fluorochromes for discriminating non-sperm events (debris): H342/SSC or H258/SSC cytograms (H342^−^ or H258^−/−^ events considered debris) for the sperm physiology analysis; AO in a red vs. green fluorescence cytogram (AO^−/−^ considered debris) for SCSA; PI/SSC or H334/SSC (H342^−^ or PI^−^ events considered debris) for the remaining techniques. In all cases, at least 5000 spermatozoa were acquired. In all cases, standard samples were run daily and after 20–30 samples for consistency between sessions (for the sperm physiology analyses, a sample with dead spermatozoa was also used as a reference). Data from the sperm physiology analysis were obtained using Weasel v. 3.7 software (Frank Battye, Melbourne, Australia); for the chromatin evaluation techniques, data was processed using scripts for the R statistical software v. 4.3 [63] and Bioconductor packages flowCore v. 2.14 and openCyto v. 1.8 [64,65].

For the sperm physiology study, the filter configuration was 450/50 nm as V1 for H342 and H258 in the violet line; 525/50 nm for YP and CFDA, 585/40 nm for M540, and 655–730 nm for PI and MX in the blue line; and 655–730 nm PNA and MT in the red line. We obtained the proportion of viable (PI^−^), apoptotic (YP^−^ as the ratio of PI^−^), damaged acrosomes (total as PNA^+^ and as the ratio of YP^−^), capacitated (MC540^+^ as ratio of YP^−^), active mitochondria (MTdr^+^), and high mitochondrial O_2_^•−^ (MX^+^ as ratio of H258^−^); the production of cytoplasmic ROS was evaluated as the median fluorescence intensity (MFI) of H258^−^ spermatozoa.

For the evaluation of 8-oxo-dG, the filter configuration was 450/50 nm in the violet line for detecting the H342 fluorescence and 525/50 nm in the blue line for detecting the FITC fluorescence (conjugated to the anti-8-oxo-dG). The levels of DNA oxidative damage were estimated from the proportion of spermatozoa with high FITC MFI.

For SCSA, AO was excited with the blue line, and the green and red fluorescences were detected with the 525/50 nm and 655/730 nm filters, respectively. The DNA fragmentation index (DFI) was calculated as the ratio of red fluorescence to total fluorescence times 1000. The spermatozoa with a high fragmentation index (%DFI) was calculated as the proportion of spermatozoa with DFI > 250, and the heterogeneity of the sperm population was calculated from the standard deviation of DFI (SD-DFI). Sperm nuclear immaturity (lower compaction) was estimated as the proportion of spermatozoa with AO green fluorescence above the 65% percentile of total fluorescence channels (high DNA stainability; %HDS).

For the mBBr analysis, the 655/730 nm filter was used in the red line for the PI fluorescence, and the 450/50 nm filter was used in the violet line for the mBBr fluorescence. The levels of disulfide bridges were estimated as the difference between sperm mBBr MFI in the DTT-treated and untreated splits, as detailed previously, divided by 2 [31]. Additionally, the sperm population was split into low, moderate, and high mBBr MFI as proportions.

For the protamination analysis, H342 and CMA3 fluorescences were assessed with the 450/50 nm filter in the violet line and the 525/50 nm filter in the blue line, respectively. The CMA3 MFI was estimated from the sperm population as an estimation of nuclear compactness. As for the mBBr analysis, we also obtained the proportion of spermatozoa with low, moderate, and high CMA3 MFI.

### 4.11. Statistical Analysis

The statistical analyses were conducted in the R statistical environment v. 4.3 [63]. The effects of the boar and storage were estimated using linear mixed-effects models, using the male and the day of preservation (D0 or D11) as fixed factors and the ejaculate as random effects. The multivariate analysis was carried out using Pearson’s correlations and variable clustering (Hubbert’s D statistic) for discovering variable associations, and principal component analysis (PCA) was used to reduce the complexity of the multiple parameters to a few variables (principal components). PCA is useful for helping understand the relationships among variables and the influence of boar and storage time on the overall chromatin structure. Results are presented as mean ± SEM except if otherwise stated. The statistical signification was set at *p* ≤ 0.05.

## 5. Conclusions

Using multiple techniques and parameters using a multivariate strategy enabled a more complete analysis of the pig sperm chromatin. Our study confirms the homogeneity of sperm chromatin in production conditions (males from a strictly managed stud center) and the resilience of sperm chromatin to prolonged cooled storage. Nevertheless, we could detect many parameters showing variability among males and sampling times, offering promise for the future development of efficient workflows for identifying suboptimal males and for quick and effective analysis of the sperm nucleus in this species. The 8-oxo-dG (optimized for its use at the cellular level in pig spermatozoa for the first time) and mBBr-derived parameters show promise for specific applications in this species. We have determined that CMA3, widely used in human spermatology, might not be appropriate for the proposed approach, but further research is needed. This methodology must be tested with heterogeneous male populations and in suboptimal situations for the stud centers to fully evaluate its sensitivity for detecting relevant changes in the sperm nucleus. Fertility data would determine the predictive power of multivariate models constructed with the tested parameters for assessing the reproductive performance of stud centers. Finally, this study highlights the relevance of chromatin analyses combined with the sensitivity and speed of flow cytometry. While flow cytometry is not routinely implemented in porcine AI centers yet, the potential for providing parameters related to the male’s reproductive performance and increasing affordability makes it a relevant technology for improving the industry’s efficacy.

## Figures and Tables

**Figure 1 ijms-25-01953-f001:**
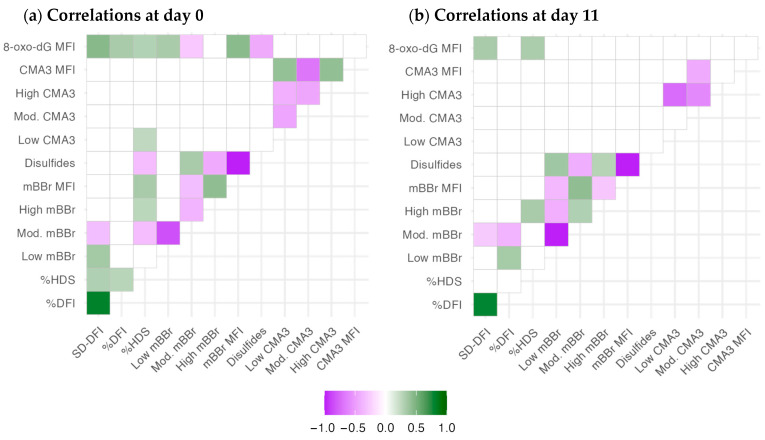
Correlation analysis for chromatin status variables at D0 and D11 of the experiment. The color gradient depicts r-values. Correlations with *p* > 0.05 (adjusted by the false discovery rate) were removed (blank spaces).

**Figure 2 ijms-25-01953-f002:**
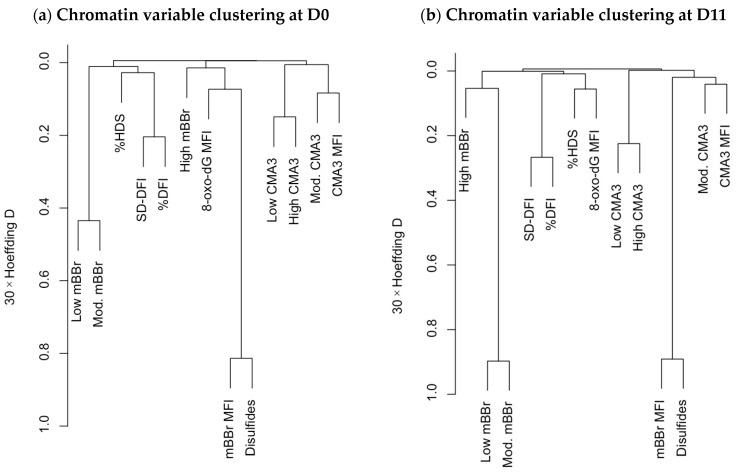
Hierarchical clustering of the chromatin status variables at D0 and D11, using the Hoeffding D statistic. Variables that are shorter distances apart and are clustered together are more closely related.

**Figure 3 ijms-25-01953-f003:**
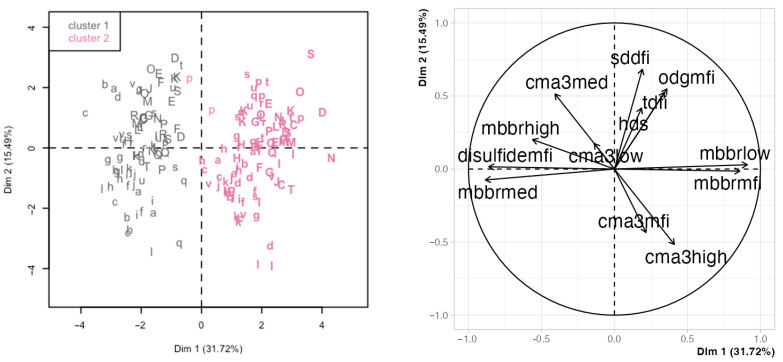
Principal component analysis (PCA) using the chromatin variables. The left panel shows the observations in the space defined by the first two principal components (PC, labeled Dim 1 and Dim 2), identified by boar (letters), day of analysis (regular font for D0 and bold for D11), and cluster (dark grey for cluster 1 and pink for cluster 2; hierarchical clustering on the PCA results). The right panel shows the variable loadings for the two first PCs (influence of the original variables on the PC). Abbreviations are defined thusly: sddfi: SD-DFI; tdfi: %DFI; hds: %HDS; mbbrlow: low mBBr; mbbrmed: moderate mBBr; mbbrhigh: high mBBr; mbbrmfi: mBBr MFI; disulfidemfi: disulfide levels; cma3low: low CMA3; cma3med: moderate CMA3; cma3high: high CMA3; cma3mfi: CMA3 MFI; odgmfi: 8-oxo-dG MFI.

**Table 1 ijms-25-01953-t001:** Descriptive statistics (median ± median absolute deviation) for the three subpopulations obtained from the motility data.

Cluster	VCL (µm/s)	VSL (µm/s)	VAP (µm/s)	LIN (%)	STR (%)	WOB (%)	ALH (µm)	BCF (Hz)	DNC (µm^2^/s)	DNCm (µm)	%
Slow	50.1 ± 22.6	14.4 ± 8.5	27.4 ± 13.2	31.1 ± 17.2	58.9 ± 23.3	57.4 ± 14.8	1.4 ± 0.4	9 ± 5.9	72 ± 51.3	4.7 ± 3.3	29.9
Fast	117.3 ± 51.8	34.5 ± 14.4	67.8 ± 26.2	32.7 ± 20.4	58.9 ± 27.7	58.2 ± 12.4	2.6 ± 1.1	20 ± 7.4	304.8 ± 238.1	8.2 ± 7.1	51.8
Hyper	113.5 ± 51.6	9.1 ± 5.8	51.7 ± 28	8.2 ± 5.8	18.5 ± 13.7	45.5 ± 9.4	2.9 ± 1.1	13 ± 7.4	331.8 ± 255.9	33.6 ± 27	18.3

VCL, curvilinear velocity; VSL, straight-line velocity; VAP, average path velocity; LIN, linearity of the curvilinear trajectory (ratio of VSL/VCL); STR, straightness; ALH, amplitude of lateral head displacement; WOB, wobble coefficient; BCF, beat cross-frequency; DNC, sperm dance; DNCm, sperm mean dance.

**Table 2 ijms-25-01953-t002:** Mean ± SEM for the sperm chromatin status parameters on each analysis day and *p* values for the factor effects (day of analysis, boar, and their interaction).

Variable	Day 0	Day 11	Day	Boar	Day × Boar
SD-DFI	15.08 ± 0.20	14.91 ± 0.20	0.997	<0.001	0.490
%DFI (%)	1.12 ± 0.07	1.41 ± 0.07	0.003	<0.001	0.468
%HDS (%)	4.31 ± 0.14	4.75 ± 0.14	0.132	<0.001	0.869
Low mBBr (%)	6.88 ± 0.59	34.18 ± 0.60	<0.001	<0.001	<0.001
Moderate mBBr (%)	88.27 ± 0.66	64.55 ± 0.67	<0.001	<0.001	<0.001
High mBBr (%)	4.84 ± 0.20	1.27 ± 0.20	<0.001	<0.001	0.005
Median mBBr (MFI)	0.95 ± 0.01	1.18 ± 0.01	<0.001	<0.001	<0.001
Disulfide levels	33.28 ± 0.31	21.31 ± 0.31	<0.001	<0.001	<0.001
Low CMA3 (%)	6.09 ± 0.20	5.66 ± 0.20	0.431	0.643	0.385
Moderate CMA3 (%)	87.46 ± 0.33	86.82 ± 0.33	0.040	0.674	0.355
High CMA3 (%)	6.45 ± 0.24	7.51 ± 0.24	<0.001	0.301	0.412
Median CMA3 (MFI)	1.083 ± 0.004	1.099 ± 0.004	0.002	0.495	0.893
8-oxo-dG MFI (%)	2.54 ± 0.05	2.80 ± 0.05	<0.001	<0.001	<0.001

SD-DFI, the standard deviation of DFI (DNA fragmentation index); %DFI, DNA fragmentation; HDS, chromatin immaturity; mBBr, monobromobimane; CMA3, chromomycin A; MFI, median fluorescent intensity for the fluorescent label.

**Table 3 ijms-25-01953-t003:** Loadings and contributions of each starting variable (chromatin status) on the two principal components computed from the PCA. The loadings correspond to the correlations of variables and components. The contributions represent the importance of each variable for each component.

Variable	Loadings		Contributions	
	PC1	PC2	PC1	PC2
SD-DFI	0.189	0.686	0.87	23.38
%DFI	0.341	0.527	2.82	13.82
%HDS	0.185	0.417	0.83	8.64
Low mBBr fluor.	0.910	0.029	20.10	0.04
Moderate mBBr fluor.	−0.884	−0.075	18.97	0.28
High mBBr fluor.	−0.562	0.202	7.65	2.03
mBBr MFI	−0.859	−0.015	17.90	0.01
Disulfide levels	−0.861	0.016	17.99	0.01
Low CMA3 fluor.	−0.139	0.169	0.47	1.42
Moderate CMA3 fluor.	−0.407	0.513	4.03	13.07
High CMA3 fluor.	0.411	−0.511	4.10	12.96
CMA3 MFI	0.216	−0.434	1.13	9.37
8-oxo-dG MFI	0.359	0.549	3.13	14.96

SD-DFI, the standard deviation of the DFI (DNA fragmentation index); %DFI, DNA fragmentation; HDS, chromatin immaturity; mBBr, monobromobimane; CMA3, chromomycin A; MFI, median fluorescent intensity for the fluorescent label.

## Data Availability

The data presented in this study are openly available in Zenodo at 10.5281/zenodo.10394482, reference number 10394483.

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
