# Peer review of "Application of Flow Cytometry Using Advanced Chromatin Analyses for Assessing Changes in Sperm Structure and DNA Integrity in a Porcine Model"

_ijms, 2024, doi:10.3390/ijms25041953_

Round 1

Reviewer 1 Report

Comments and Suggestions for Authors

The manuscript discusses changes in sperm structure and dna integrity by assessing chromatin, using pigs as the study model.

Major issues

-Do you have proof that the 36 boars had actually sired any offspring?

-How were the doses from these 36 animals selected?

-Please include the scree plots for the two PCAs.

-The PCA revealed <50% importance of the two principal components, so please comment on that.

-The Discussion is extremely difficult to read, so please re-organise, separate in sub-sections and please use shorter sentences.

-Please add a sub-section about the clinical significance of the findings.

Minor issues

-The Introduction is on the longer side and possible some passages can be deleted or else transferred to Discussion.

-The paragraph with the objectives of the study should be better phrased.

-Table 2 may be moved to supplementary material.

-Some references from 2022-23 relevant to this study have been missed.

Author Response

The manuscript discusses changes in sperm structure and dna integrity by assessing chromatin, using pigs as the study model.

> We thank the reviewer for their valuable comments on the manuscript.

Major issues

-Do you have proof that the 36 boars had actually sired any offspring?

> Yes. The boars were selected to enter the stud center with the sole aim of producing semen doses for artificial insemination. These centers have strict semen quality control, and the genetics company (Topigs-Norsvin) continuously receives fertility data from the sow farms. Any deviation is immediately detected, and any sign of subfertility implies removing the boar from the stud center. The center has indicated that no incidences were reported regarding the boars producing the semen doses for the study. We have added some sentences to 4.2 on this regard.

-How were the doses from these 36 animals selected?

> These animals were randomly selected from the ~120 boars producing semen doses in the stud farm (as indicated above, all tested semen quality and regularly tested for fertility data). The only constraint was the age (between 12 and 22 months of age). Animals enter the center at ~6 months of age; these young animals are more variable and, in some cases, have lower sperm quality; boars start being removed after 1 year of AI dose production. Therefore, these animals' age range ensures consistency and is limited by the removal from the center (still with good semen quality).

-Please include the scree plots for the two PCAs.

> We have added them to the supplementary material. We have also included more information about the PCA for the chromatin analysis. In the new Fig. S6, we included all the combinations of the PC above an eigenvalue of 1 (four selected by that criteria), except for the first two, shown in the manuscript. We focused on these first two (essentially, the first one was the most informative) to avoid adding too much complexity to the manuscript. Nevertheless, we agree that this extra information is interesting to describe our results thoroughly, and we have added comments in the results section. Regarding the PCA from the complete set of variables, the results show that the sperm motility and physiology variables could be less useful, at least separating both days of analysis, and they could add noise to the PCA.

-The PCA revealed <50% importance of the two principal components, so please comment on that.

> Added. Since we started from a lot of variables, the first PC contained only part of the variance, although they were illustrative of the data spread and the influence of the individual variables. As commented above, we have included extra information in the supplementary material. We also consider this a good point to clarify, and we have revised the results to comment on that.

-The Discussion is extremely difficult to read, so please re-organise, separate in sub-sections and please use shorter sentences.

> Thank you. We agree that it became difficult to follow due to the number of variables and the discussion of implications. We have amended the discussion and organized it around sub-sections.

-Please add a sub-section about the clinical significance of the findings.

> We have expanded the discussion on that aspect.

Minor issues

-The Introduction is on the longer side and possible some passages can be deleted or else transferred to Discussion.

> We agree. We have consolidated part of it in the discussion while following the previous recommendations.

-The paragraph with the objectives of the study should be better phrased.

> Revised; thank you for the comment.

-Table 2 may be moved to supplementary material.

> We have split in two, keeping the information on the chromatin parameters in the main text. We agree that Table 2 was excessively large. However, since we are focusing on the analysis of sperm chromatin, we believe it could be helpful for the reader to have that data in the main text, with the rest of the variables being described in the supplementary material.

-Some references from 2022-23 relevant to this study have been missed.

> We have revised the most current bibliography and added some references.

Reviewer 2 Report

Comments and Suggestions for Authors

The authors conducted an excellent and comprehensive study on the chromatin status, and the changes induced during the refrigeration period.

The manuscript is well-written, and the results obtained relevant; nevertheless, there are some aspects that authors should address before the acceptance of the manuscript for publication.

Line 498. Authors should specify if the commercial semen extender used to obtain AI doses is either a long-term extender or an extra-long-term extender. Considering that the refrigeration period is of 11 days, this is a relevant point to address.

Lines 503-504. Authors analysed the sperm sample at day 0 and after 11 days of refrigeration. Authors should clearly specify why did they analyse only one temporal point, and they did not intermediate temporal points during the refrigeration period.

In the Conclusion section authors should emphasise the applied interest of the study. Considering that flow cytometry is not routinely used in pig industry, authors should also provide some clues of how AI centres could incorporate the analysis of these parameters to assess the sperm quality of fresh and refrigerated ejaculates.

Comments on the Quality of English Language

The manuscript is well-written, and it only contains some minor typographical errors that must be addressed during the revision process.

Author Response

The authors conducted an excellent and comprehensive study on the chromatin status, and the changes induced during the refrigeration period.

The manuscript is well-written, and the results obtained relevant; nevertheless, there are some aspects that authors should address before the acceptance of the manuscript for publication.

> We thank the reviewer for their comments. We have revised the text for language mistakes, too.

Line 498. Authors should specify if the commercial semen extender used to obtain AI doses is either a long-term extender or an extra-long-term extender. Considering that the refrigeration period is of 11 days, this is a relevant point to address.

> We agree this is a significant omission. We have added it to the text.

Lines 503-504. Authors analysed the sperm sample at day 0 and after 11 days of refrigeration. Authors should clearly specify why did they analyse only one temporal point, and they did not intermediate temporal points during the refrigeration period.

> We have added that information to the text.

In the Conclusion section, the authors should emphasise the applied interest of the study. Considering that flow cytometry is not routinely used in pig industry, authors should also provide some clues of how AI centres could incorporate the analysis of these parameters to assess the sperm quality of fresh and refrigerated ejaculates.

> Thank you. This advice is very appropriate, since that is the ultimate goal of the project we are carrying out, and of which this manuscript is the first report. We added it to the conclusions.

Reviewer 3 Report

Comments and Suggestions for Authors

Dear Authors,

In my opinion, the reviewed paper is innovative and interesting, as it presents the possibility of using and comparing many modern laboratory techniques to assess the state of nuclear chromatin in stored domestic pig sperm. In addition to the commonly used SCSA test, the studies used less frequently used tests such as 8 oxo-dG and mBBR and CMA-3 to assess DNA damage. The study obtained interesting results indicating that the CMA-3 test used in the analysis of the nuclear chromatin state of human sperm does not seem to be suitable for boar sperm. In addition, the study also carried out comprehensive studies of other cellular structures of preserved sperm and examined the relationships between the examined parameters assessing the status of nuclear chromatin, which deserves special mention.

The laboratory analyzes were performed in accordance with the previously provided protocols and do not raise any objections.

The analysis of the results is also carried out properly, although the authors also introduced elements of discussion that I would suggest removing from this chapter of the work (e.g. lines 141-148).

A minor remark concerns subsection 4.5. in the methodology (line 519), which should be moved below.

The discussion is interesting, but very extensive and could be limited.

Author Response

In my opinion, the reviewed paper is innovative and interesting, as it presents the possibility of using and comparing many modern laboratory techniques to assess the state of nuclear chromatin in stored domestic pig sperm. In addition to the commonly used SCSA test, the studies used less frequently used tests such as 8 oxo-dG and mBBR and CMA-3 to assess DNA damage. The study obtained interesting results indicating that the CMA-3 test used in the analysis of the nuclear chromatin state of human sperm does not seem to be suitable for boar sperm. In addition, the study also carried out comprehensive studies of other cellular structures of preserved sperm and examined the relationships between the examined parameters assessing the status of nuclear chromatin, which deserves special mention.

The laboratory analyzes were performed in accordance with the previously provided protocols and do not raise any objections.

> We thank the reviewer’s comments.

The analysis of the results is also carried out properly, although the authors also introduced elements of discussion that I would suggest removing from this chapter of the work (e.g. lines 141-148).

> We agree that they could be excessive in that part. According to the advice, we have revised this section.

A minor remark concerns subsection 4.5. in the methodology (line 519), which should be moved below.

> Thank you, corrected.

The discussion is interesting, but very extensive and could be limited.

> We have also revised and reorganised the discussion according to the other reviewers’ comments. We shortened it, and it might be clearer now.

Round 2

Reviewer 1 Report

Comments and Suggestions for Authors

The authors did not answer the following query.

-Do you have proof that the 36 boars had actually sired any offspring?

The response of the authors is indirect and does not answer the question, to the point that it raises suspicions that the true answer is NO.
A positive answer to the question can only be the following: "Yes, we have seen piglets born from sows, 110-118 days after insemination with semen from these boars and these sows had indeed farrowed these piglets whilst no subsequent mating or other insemination had occurred.".
Any other answer is really misleading. So, a revised version taking of the manuscript into account the above, will be welcome.

Author Response

Thank you for the correction. Indeed, sows artificially inseminated with doses from the boars of the study farrowed at the specified time. As the reviewer indicated, the sows were inseminated once, no boars were in contact with the sows, and parity was high and within the farm's normal range.
The Quality Director of Topigs-Norsvin in Spain provided this information, which ensured that the boars in this study showed excellent fertility results (data from dozens of client farms and hundreds of AI).
The authors have requested a letter from the company confirming that information. It is attached to this reply.

Round 3

Reviewer 1 Report

Comments and Suggestions for Authors

No further comments.